# An NMR Database for Organic and Organometallic Compounds

**DOI:** 10.3390/molecules30163442

**Published:** 2025-08-21

**Authors:** Stefan Kuhn, Markus Fischer, Herman Rull, Christophe Farès

**Affiliations:** 1Institute of Computer Science, University of Tartu, Narva mnt. 18, 51009 Tartu, Estonia; rullherman@gmail.com; 2Institute for Medical Physics and Biophysics, Leipzig University, Härtelstr. 16-18, 04107 Leipzig, Germany; markus.fischer@medizin.uni-leipzig.de; 3Max-Planck-Institut für Kohlenforschung, Kaiser-Wilhelm Platz 1, 45470 Mülheim an der Ruhr, Germany; fares@mpi-muelheim.mpg.de

**Keywords:** chemical information, compound database, organometallic compound, NMR

## Abstract

Databases of organic compounds are rooted in chemical formats, typically modeling chemical structures as simple graphical representations, which are easily readable by both humans and machines. This information can be handled by software and standards that have emerged in chemistry over decades and are well established. The present communication proposes that the extension of databases to encompass organometallic compounds, with coordination bonds, can be accomplished seamlessly with minor modifications, without compromising the utility of the database functionalities. Using a database for nuclear magnetic resonance (NMR) data, we demonstrate that this enables treating the organometallic compounds with the same algorithms as the organic compounds.

## 1. Introduction

Chemical structure representation is a problem as old as the concept of molecules, going back to John Dalton and the early 1800s [1]. For organic molecules, standards emerged over time and are published in [2] (now online at https://iupac.qmul.ac.uk/drawing/drawing.html (accessed on 19 February 2025)). Apart from some marginal areas, chemists usually can agree on what a good or correct depiction is. Those drawings normally consist of covalent bonds (single, double, triple) linking atoms, potentially annotated with charge or isotope information. For the sake of simplicity, the use of carbons and hydrogens has become optionally implicit. There are sometimes different representations for aromaticity or stereochemistry, but overall the consensus is high.

This changes for organometallic compounds or coordination compounds. Here, a wide variety of potential depictions are possible, and there is no agreement on which is best. Table 1 shows a number of ways to draw ferrocene.

As long as all depictions are handled manually, differences are not too problematic, since expert readers can mentally “translate” depictions as necessary. When electronic resources are involved, the problem becomes more relevant: computers need explicit information and are easily confused by issues which have been silently corrected by humans so far. Furthermore, chemical information and modeling are built upon the standard depictions of organic molecules. A classic example is the molfile format [3], which has been for decades and probably still is the standard way to save and transmit molecular information. This format treats molecules as graphs consisting of atoms as vertices and bonds as edges. Other chemical file formats are built on similar principles; see [4] for an overview. In such file formats, encoding a multi-center bond (Table 1e) is not compatible, since the bond that connects the iron atom to the ring does not end in an atom. However, beyond chemical depiction, the “atom–bond–atom” representation is at the heart of graph algorithms. Graph algorithms are an area extensively treated in computer science and are used for chemical problems, such as substructure search [5,6].

**Table 1 molecules-30-03442-t001:** Different ways to depict a ferrocene compound. Only one cyclopentadienyl ring is shown, the other ring would be represented the same way. Meaning of symbols: ✓: Correctly handled, (✓) partly correctly handled, □: Not handled. Partly from [7].

	a	b	c	d	e
	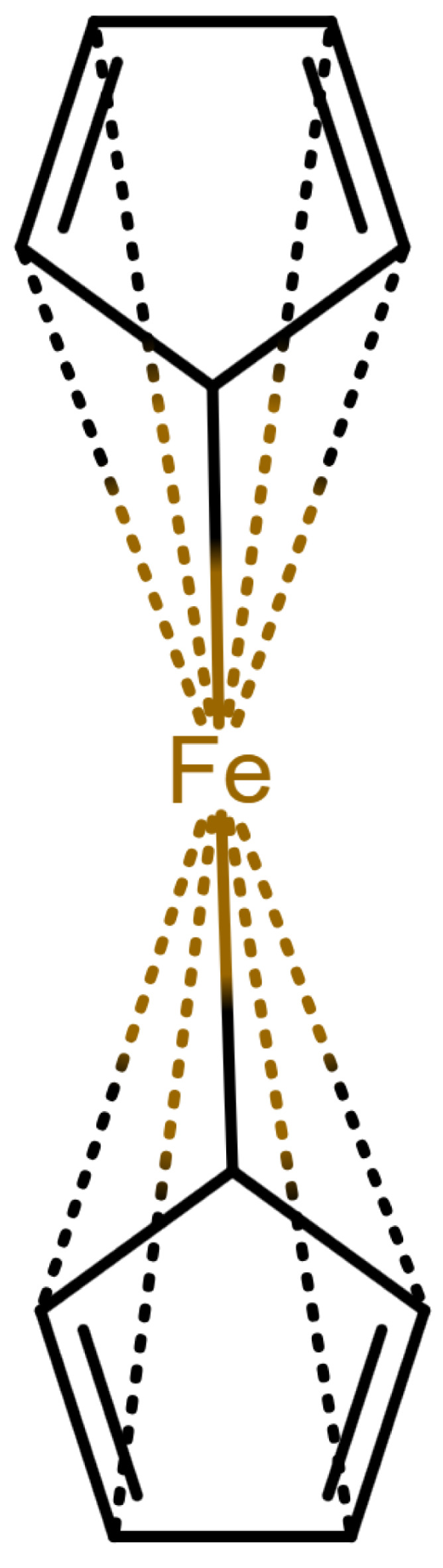	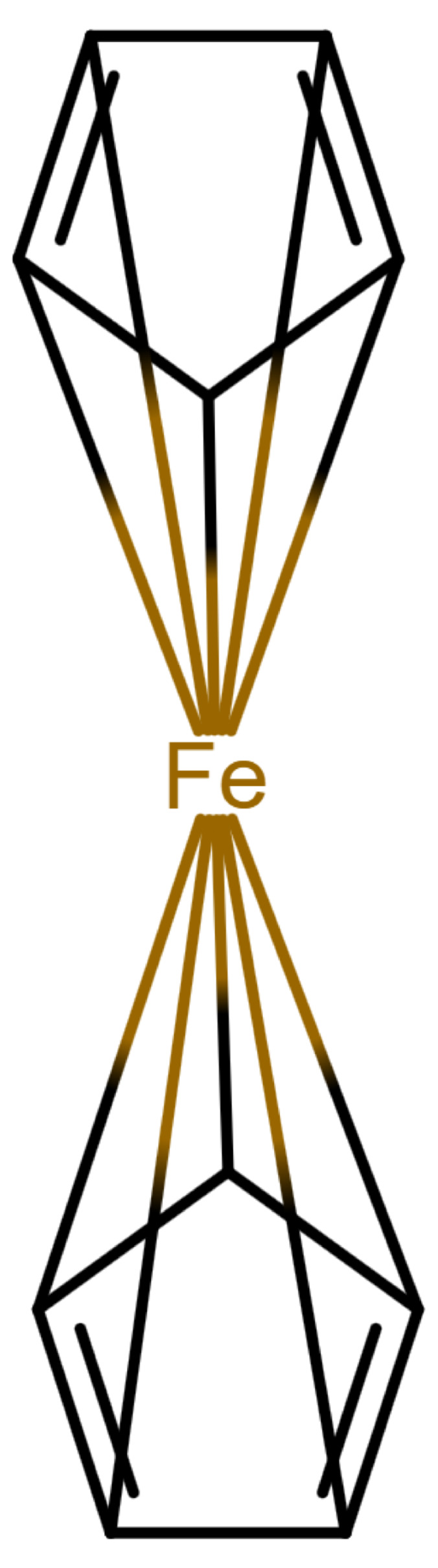	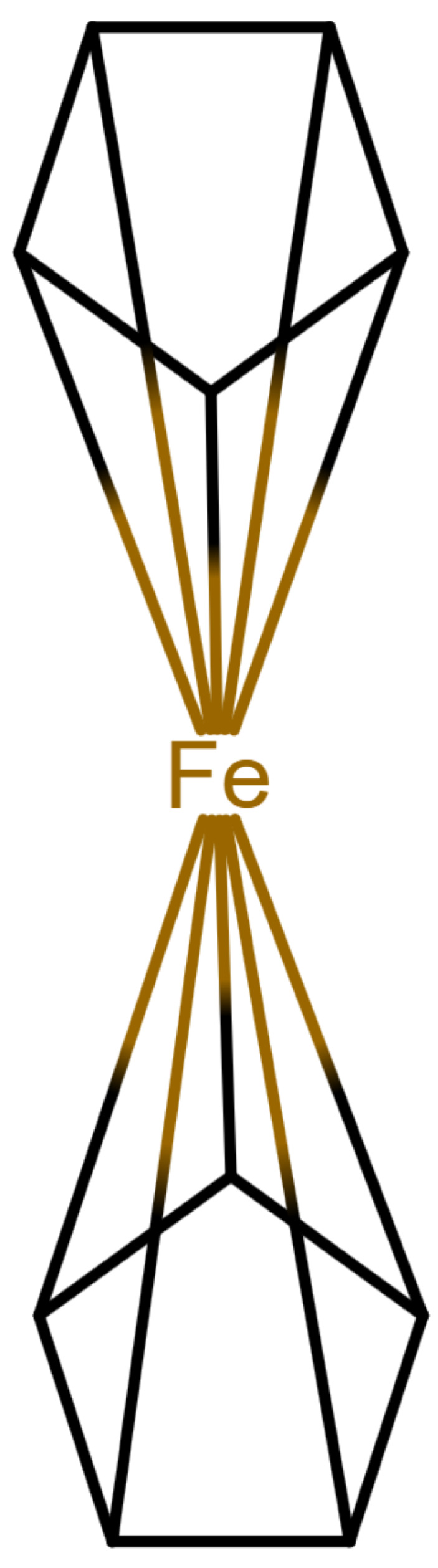	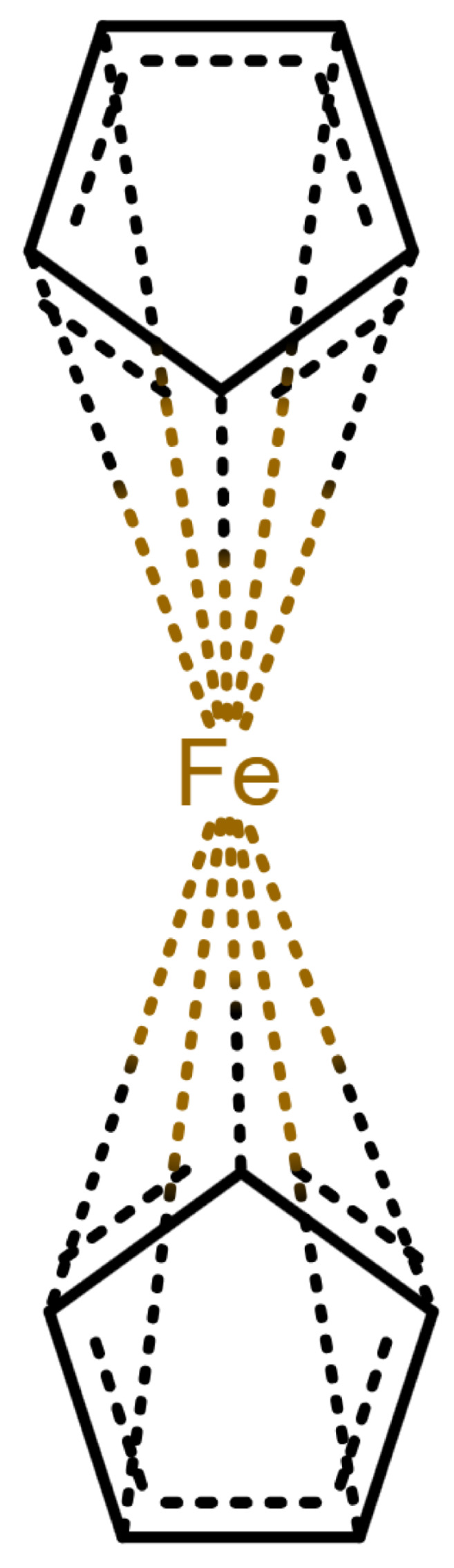	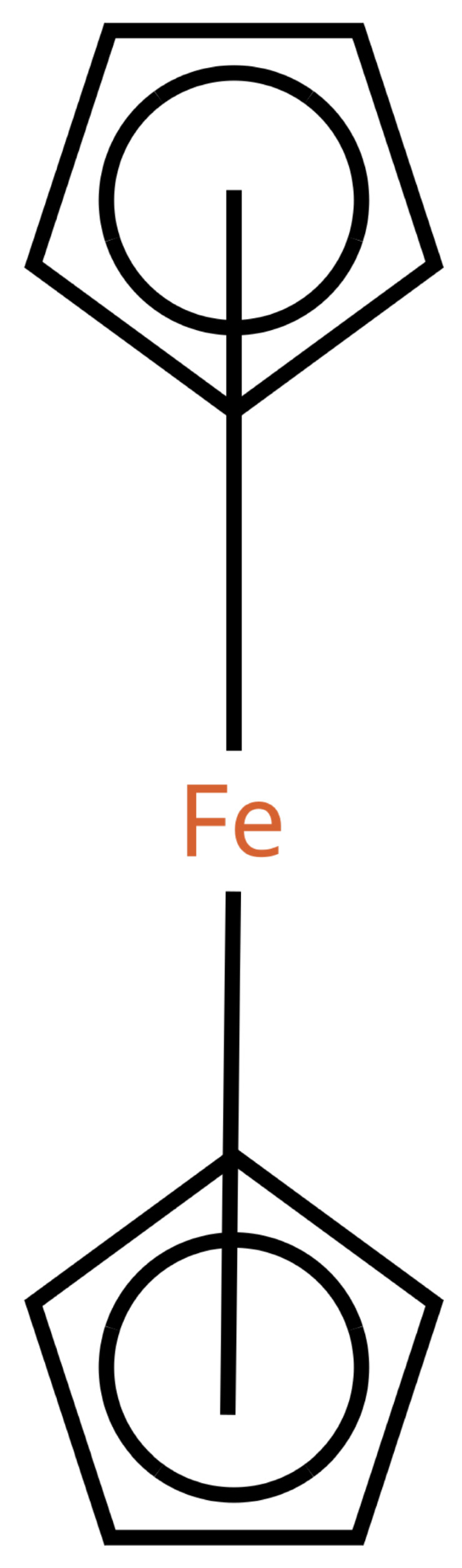
Valence	✓	□	✓	✓	✓
Equivalence	□	□	✓	✓	✓
Aromaticity	✓	✓	□	✓	✓
Simplicity	✓	✓	✓	□	✓
molfile compatibility	( ✓)	✓	✓	(✓)	□

Chemical databases, when holding structure data, mostly follow the graph-based approach. A typical example is the BIOVIA Direct cartridge for Oracle databases [8], which originated as a product of MDL Information Systems. It saves graphs into the database, but allows searching, reading, and writing molecules in a variety of formats. Whilst MDL was a pioneer in the market, there are now a multitude of similar systems available. For an overview, see [9]. There are also databases which contain crystal structures, e.g., CSD [10], OCD [11], or OMDB [12]. These also contain organometallic compounds, but since the file format used, Crystallographic Information File (CIF), does not contain bonds, the problem of depiction does not arise. On the other hand, in many applications, bonds are needed, e.g., for substructure searches or as input for machine learning. The same is true for Ligand Knowledge Bases, e.g., [13], which publishes structures as XYZ files, also not containing bonds. The Reaxys database, a well-established database for organic chemistry, allows searching for organometallic substances. In structure searches, coordination bonds must be drawn as single bonds [14].

Here, we use the nmrshiftdb2 database as an example. In it, molecules are also saved as graphs, but with separate tables for atoms and bonds. Because of this, atoms have IDs on the database level, and shift assignment can be performed using database IDs, allowing the database to check referential integrity. For atoms and bonds, the same information is saved as can be found in molecular drawings, including wedge bonds (see [15] for details).

The handling of coordinative bonds in this work is based on [7]. This paper suggests a chemical file format, based on molfiles, extended by two features: A zero-order bond (or, for short, zero bond) and an explicit hydrogen count. A zero bond, which is always between two atoms, is used for “any bond that is not a well-defined covalent bond” [7]. Figure 1 shows a ferrocene drawn with zero bonds. Second, there is an option to specify the hydrogen count of an atom. Automatic calculations are only carried out for C, N, P, O, or S atoms. A similar suggestion as in [7] was made in [16], but this is restricted to drawings only and does not mention file formats.

The depiction in Figure 1 has at least the following advantages:The correct number of hydrogens is given or can be determined by rules.The π character of the ring and therefore aromaticity is visible.The oxidation state of the metal can be inferred correctly.All significant interactions are represented.The structure depiction forms a graph.

Looking at Table 1, the format ticks most of the boxes. It does not represent the five atoms in the rings as equivalent, but we see in Section 3.4 that this is not a major issue. We mark it as “conditionally molfile compatible” since a minor extension of the molfile standard is needed.

In Table 2, we show for a number of different types of coordination bonds how they are treated in this system. For coordination complexes, it is interesting to note that the depiction does not show the location of the atoms by wedge bonds. This is primarily due to the non-existence of “zero-order wedge bonds”. Interestingly, mol files have the possibility to accommodate this, since the bond order and the stereo property are different columns in the bond block [3] (p. 50). The use of the stereo property is restricted to single bonds, but it would not break the format to use them with the zero bond order.

The nmrshiftdb2 database [17] is an open-access database of NMR spectra. It includes raw data, but most prominently assigned peak lists. Technically, all data are recorded in a relational database (MySQL, but other products would work as well). In the database, molecules are recorded as graphs, which enables recording of shift assignment on the database level. This principle has worked well to maintain consistency and enable searches and other functions (e.g., shift prediction, where the assignment is crucial). We can easily add organometallic atoms, represented using the zero-bonds, by just allowing an additional bond type (all hydrogen atoms are explicitly recorded anyway). As we show in Section 3, the functionalities of the database nicely work on those extended data. Since nmrshiftdb2 uses the Chemistry Development Kit [18] for many functions (e.g., generating images or performing substructure searches), some changes to the CDK were necessary as well.

In the context of NMR databases, an additional advantage of zero-bonds is that J-couplings only occur between atoms sharing electrons, directly or indirectly. This means that is easy to know where coupling constants could be expected to be entered or not.

Looking at other databases, it is obvious that a uniform way of handling organometallic structures does not exist. Figure 2 shows the representation of Ferrocene in ChemSpider. Clearly, this is not an ideal depiction since the iron atom seemingly forms no connections to the rings at all and the charge distribution suggested is not realistic. Also, we note that the InChI and the SMILES generated include the Fe atom as a separate atom, which is not realistic. ChemSpider developers discuss the issues in https://blogs.rsc.org/chemspider/category/cheminformatics/?doing_wp_cron=1739375607.6039280891418457031250 (accessed on 20 February 2025), Section “Organometallics”. Figure 3 shows the two entries for Ferrocene shown in the NIST Chemistry WebBook. They have different depictions, but the same InchI (which is the same is in ChemSpider), so there must be something else used for deciding structure identity. Both representations are not ideal, and having two entries (linking to each other) is problematic.

A database specifically for ^29^*Si* data is the database in [19]. As shown in Figure 4, it uses a type of zero bond, although it seems that this is not used uniformly and is not explained. Finally, nmrsharc [20] is a project to archive files from ab initio calculations. This can include organometallic and inorganic molecules, but the project extracts the protocol files from the quantum chemistry software and does not deal with display or search questions.

In this work, it is demonstrated that the method of encoding non-standard metal-coordinated molecules in the mol-format, as outlined in [7], facilitates the construction of a database comprising organic, organometallic, and coordination compounds using algorithms designed for organic molecules. Additionally, valuable results can be obtained from this database with minimal additional coding. Our demonstration case is an NMR database that holds all mentioned types of molecules and offers functionalities like NMR chemical shift predictions. We are aware that proper handling of metal compounds in general in databases would be desirable. We consider this work as a stepping stone and a contribution to the development of a final solution.

A deeper problem, which we cannot solve here, is that there is often a confusion between the graphical presentation of a chemical structure and the computational storage of it. Both could in theory be independent, but in practice, there is an implicit tendency to make them align where possible. An example for the confusion of this is the specification of the bond block of the mol file (Figure 5). On the one hand, the numbers which are normally translated to wedge bonds are described as “bond stereo, 0 = not stereo, 1 …”, not mentioning wedge bonds at all. On the other hand, the notes column says “The wedge (pointed) end …”, implying wedge bonds, which are not mentioned anywhere else in the document. We follow the approach to keep visual and computational representation in line where possible here, but we are aware of the deeper problems.

## 2. Results and Discussion

Using zero bonds, it is possible to consistently save organic and organometallic molecules in a single database. Structures, including organometallic compounds, in this database can be saved and retrieved by a variety of searches, including substructure search. It is also possible to perform NMR shift predictions using HOSE codes.

Looking back at Table 1, we can see that this is not the case all representations. For representation (e), building a HOSE code would be difficult due to the lack of connections. For (c), the lack of aromaticity would prevent building a HOSE code containing aromaticity, which is an essential feature. For (b), the HOSE code could work, given that the aromaticity could be detected correctly, which is difficult due to the incorrect valence of the ring carbons. For (c), a HOSE code would work, but the display is very unusual for chemists otherwise. Overall, the zero-bond-based depiction is a good option.

The need to do some extra treatment for equivalence of the ring atoms could be considered a drawback at first sight. On the other hand, this is a logical extension of how aromatic rings are treated in the standard HOSE code, making it a natural step.

Some changes to the nmrshiftdb2 software as well as to CDK needed were necessary to achieve this. Those changes were relatively easy to make and bring about a significant extension of the database. To our knowledge, nmrshiftdb2 is the first NMR database for organic and organometallic compounds, enabling consistent handling and prediction for both. The inclusion of HOSE code prediction shows that NMR-specific functionalities can work for organometallic compounds using zero bonds.

A major application of the system is NMR shift prediction. In order to demonstrate this, we show a number of examples in Table 3. We give the predicted shift for all carbon and connected hydrogen atoms, the real shifts, and the errors. All examples are calculated ignoring the structure to predict in the database, since the HOSE code algorithm used always predicts a structure from itself if it is in the database. In addition, it is important that the results depend very much on the similarity of the entries in the database, so the quality of organometallic predictions depends strongly on the coverage of the organometallic compounds in the database, just as with other fields of chemistry. It is clear from the results that more data are needed and more advanced prediction methods should be tested.

## 3. Materials and Methods

The extended nmrshiftdb2 software V. 1.5 implements the following possibilities.

### 3.1. Structure Input, Handling, and Display

A necessary tool for the database is the ability to enter structures. We use Ketcher [21] for this. Ketcher is based on JavaScript, can run in all relevant browsers, offers a wide range of features, and is open-source under the Apache License, Version 2. These properties make it suitable for nmrshiftdb2 in general. Ketcher does not offer the option to enter zero bonds as defined in [7]. We therefore use the (visually similar) “any bond” search bond type, which is possible in Ketcher, as a replacement at drawing time. In the database, they are saved as zero bonds and are shown and exported as such. In search inputs, “any bond” search bonds are searched as zero bonds. Clearly, “any bond” search bonds and zero bonds are not the same, but there is, to our knowledge, currently no tool allowing the input of zero bonds.

In order to handle the zero bonds in the CDK, we extend the enum IBond.Order by a type ZERO. For algorithms that calculate valences, this has the effect of the zero bond not contributing anything, as intended. Further changes in the CDK include the display of the zero bonds as dashed lines. Since the images of structures shown in nmrshiftdb2 are all generated using CDK, this ensures proper display of structures.

CDK offers algorithms to saturate molecules by adding missing hydrogens. This in turn relies on atom typing, which is not working with some metals and the zero bonds. We therefore do not use CDK for these purposes, but follow [7] by only calculating hydrogens for C, N, P, O, or S atoms. For all other atoms, hydrogens need to be entered by the user explicitly.

### 3.2. Structure Search

nmrshiftdb2 is a compound-oriented database, which has one entry for every chemical compound (but potentially multiple spectra). For this, an unambiguous identification of compounds is needed. After using the stereo-chemistry aware implementation of the CDK for several years, we switched to the International Chemical Identifier (InChI, [22]) once this became available, since it provides the most thorough approach. Specifically, we use the InChI with a fixed H layer. This is because NMR can distinguish, e.g., tautomerism [23] and the standard InChI records all tautomers as identical, whereas with the fixed H layer it can capture the tautomers. This has served the purpose of structure identification very well.

On the other hand, the InChi cannot currently handle organometallic compounds. There is an InChI project for this (https://iupac.org/project/2009-040-2-800/ (accessed on 3 February 2025)), but its status is unclear (https://web.archive.org/web/20240616134141/https://www.inchi-trust.org/organometallics/ (accessed on 10 August 2025)). We therefore decided to use TUCAN [24] instead of the InChI. We still use the InChI for purely organic compounds since it is the most widely used solution.

Since the TUCAN uniquely identifies an organometallic compound, we use it also for structure searching. By generating the TUCAN and looking for it in the database, compounds can be found reliably. A consequence of this is that the TUCAN code is the standard for defining which compounds are identical. Since TUCAN is made for this purpose, this is an intended effect. For the structure input, we handle hydrogens as before, so hydrogens are added automatically to C, N, P, O, or S atoms only.

Since we keep the InChI for non-organometallic compounds, there is a need to decide which category the input belongs to. The rule used is that a compound is organometallic if it contains at least one zero bond and a metal from a predefined list. Otherwise, the InChI is used for searches.

We also save SMILES strings [25] in the database for informational purpose. For this, we extend the CDK SMILES generator to include zero bonds. They are encoded as ’, which is a symbol not used in SMILES. The symbols .-=#$ are used in SMILES for bonds with increasing bond orders, where . means no bond, i.e., separate structures. This becomes .’-=#$ with our symbol, roughly preserving the idea of “light” to “heavy” symbol.

### 3.3. Substructure and Similarity Search

nmrshiftdb2 enables search by substructure. This returns any molecule which contains the given molecule as a substructure, or, to put it differently, it returns all superstructures of the search structure. For defining what a substructure is, the atom (symbol, charge) and bond attributes (bond order) are used. The substructure search is performed using a two-step process and implemented by CDK tools. The first step is a pre-filtering step and uses a list of bits, called a fingerprint, where each bit indicates the presence of a certain feature in the structure. If and only if the fingerprint of the search structure combined by a bitwise and function with the fingerprint of the candidate structure equals the fingerprint of the search structure, a structure can be a superstructure of the search structure. Mathematically, this means(1)G substructure of H⇒fp(G)∪fp(H)=fp(G)
where G,H are molecules, fp is the fingerprint function, and ∪ the bitwise and-function. Since structures can be found that are not actual superstructures (the inverse of Equation (Equation 1) does not hold), in a second step, a precise substructure matching algorithm, implemented in the UniversalIsomorphismTester class of CDK, is used. Using statistical terminology, it could be said that the first step can produce false positives, but no false negatives, and the second step removes the false positives. The two-step process is efficient, since the fingerprint search (performed directly in the database) is significantly faster than the exact search, and the process minimized the number of structures for the exact search. In nmrshiftdb2, there is a cut-off after 300 hits since more results are not realistically viewed.

The fingerprint algorithm would work unchanged for the organometallic compounds, but we decided to extend the fingerprint by two bits: One to designate the presence of at least one zero bond in the molecule and one to designate the presence of one of the metals Hf, Co, Zr, W, Ni, Pt, Os, W, Ti, Ni, Rh, Fe in the molecules. Those bits ensure that the fingerprinting step with organometallic fragments is not returning many false positives. For example, a substructure search for ferrocene at the time of writing ultimately yields 12 results. This is actually the result of the first step as well, showing that this is very precise and making the overall search practically instantaneous.

For the exact substructure match, we modified the UniversalIsomorphismTester class in CDK to match zero bonds with zero bonds only. Since we have no other changes, structures need to be entered in the correct way, which is intended.

### 3.4. Prediction

Using the zero bonds, it is possible to predict NMR shift values for organometallic compounds using the same tools as for organic compounds. In particular, it is possible to use HOSE (hierarchically ordered spherical environment) codes [26] for this purpose. HOSE codes encode the environment of an atom in a sphere-wise manor, allowing to search for atoms with a similar environment by searching for HOSE codes, which are text and can easily be searched in a database. The shifts of the atoms with the most similar atoms form the prediction for an atom in an unrecorded molecule. For a detailled explanation, see [27].

We extended the original HOSE code by the < symbol for zero bond (HOSE codes use > for coordinate bonds, which is not the same as a zero bond, but a related concept). Zero bonds are then followed like any other bond when building HOSE codes. The result is a HOSE code which can predict shifts from similar structures just like in organic molecules.

Figure 6 shows an example of a ^13^*C* NMR prediction for an unknown ferrocene compound. Other ferrocene compounds are contained in the database and the values of those are used. For Atoms 1 to 5, the prediction uses four spheres, for Atoms 9 and 10 as well, but only three spheres for 6 and 8 and only two for Atom 7 (hence the warning). Since the ferrocene compounds in the database have a different group attached to the ring, there are less useful data the closer we get to it.

It should be noted that for the ring Atoms 1 to 5, the HOSE code is identical. This corresponds to the chemical meaning of the representation. In order to achieve this, the single bond from Atom 1 to the Fe atom is considered as a zero bond for the HOSE code (we consider the five ring atoms equivalent). The double bonds in the ring are not encoded since the aromatic nature is enough to encode the ring. This encoding is not novel, it is used in the standard HOSE code as well. For example, the HOSE codes of the carbons in a pyrolle molecule are encoded independently of the position of the double bonds and hydrogen because the double bonds in aromatic rings and hydrogen atoms are ignored by default. Otherwise, the depiction would influence the HOSE code and the prediction. We use the same concept to ensure correct predictions for organometallic compounds.

### 3.5. Submission

A manual submission of an organometallic compound uses the same user interface as any other submission. The handling of organometallic structures is the same as in other structure inputs; the user needs to enter the “any bond” search bond, which is treated as zero bonds. Shift assignment and all other steps are as normal.

In line with nmrshiftdb2 policies, all entries are manually reviewed. During the review, equivalent atoms (for example, the ring atoms in ferrocene mentioned in Section 3.4) are also displayed and approved as part of the submission.

## 4. Conclusions

We used zero bonds and explicit hydrogen counting to extend the nmrshiftdb2 database to hold organometallic and coordination compounds in addition to organic molecules. This was possible with only a few changes to the codebase. We demonstrated that the database can perform structure searches, including substructure search. In the future, we will add more data to the database, including shift values for metals, and we will employ advanced prediction techniques for these. We will investigate whether the representation works well, for example, for graph neural networks.

## Figures and Tables

**Figure 1 molecules-30-03442-f001:**
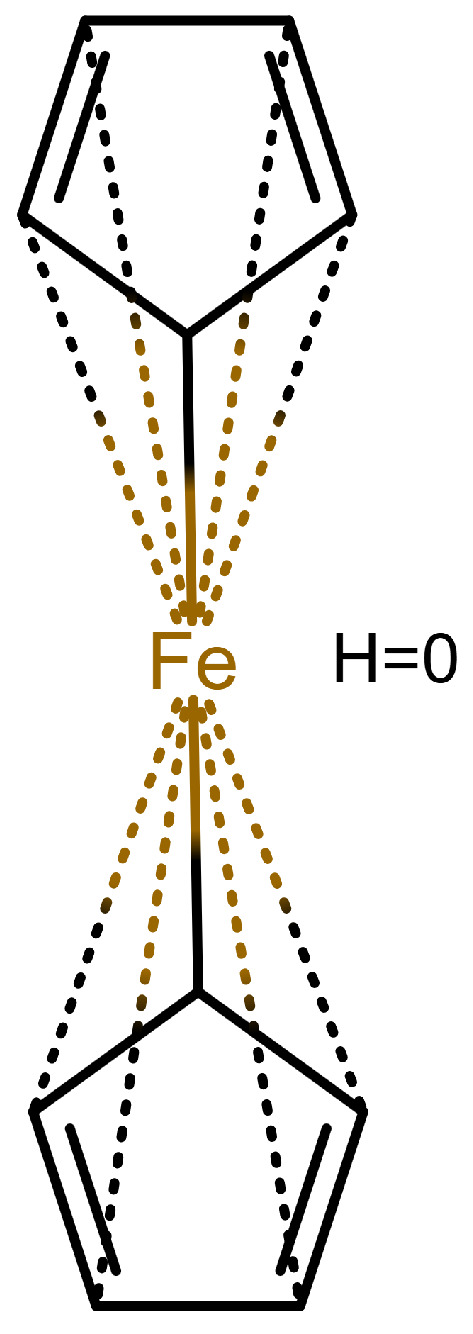
Ferrocene depicted using zero bonds, following [7]. Dashed bonds represent zero bonds. The Fe atom has an explicit hydrogen count.

**Figure 2 molecules-30-03442-f002:**
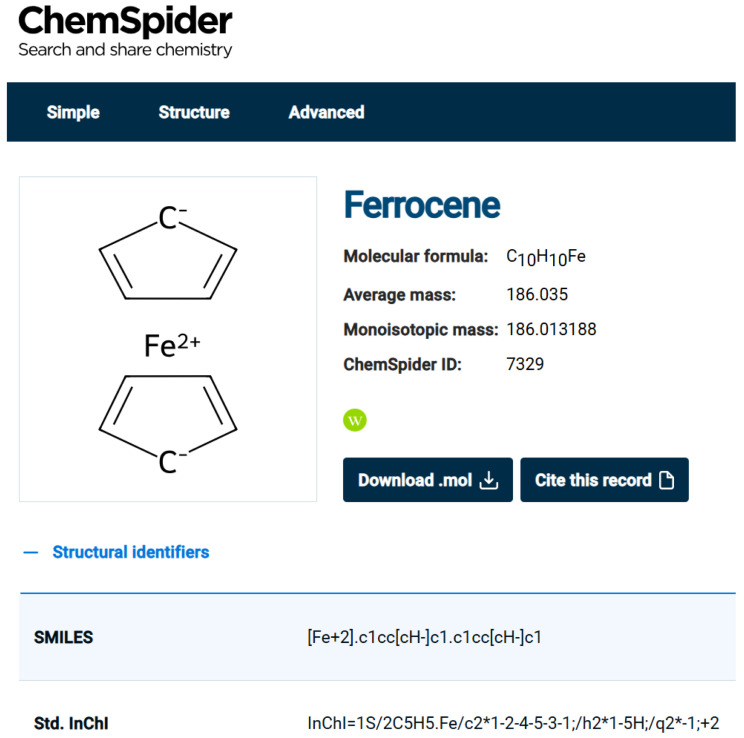
Ferrocene, as represented in ChemSpider (https://www.chemspider.com/Chemical-Structure.7329.html (accessed on 20 February 2025)).

**Figure 3 molecules-30-03442-f003:**
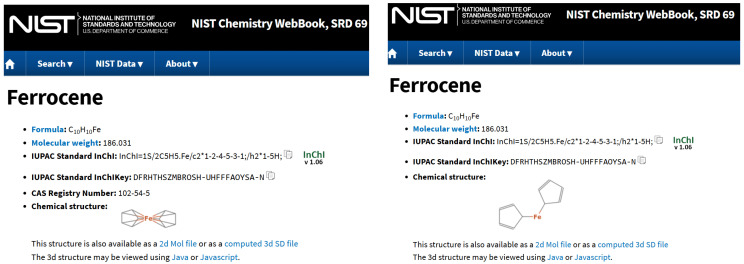
Ferrocene, as represented in the NIST Chemistry WebBook (https://webbook.nist.gov/cgi/cbook.cgi?ID=C102545 (accessed on 20 February 2025), (**left**), and https://webbook.nist.gov/cgi/cbook.cgi?ID=B6000428 (accessed on 20 February 2025), (**right**)).

**Figure 4 molecules-30-03442-f004:**
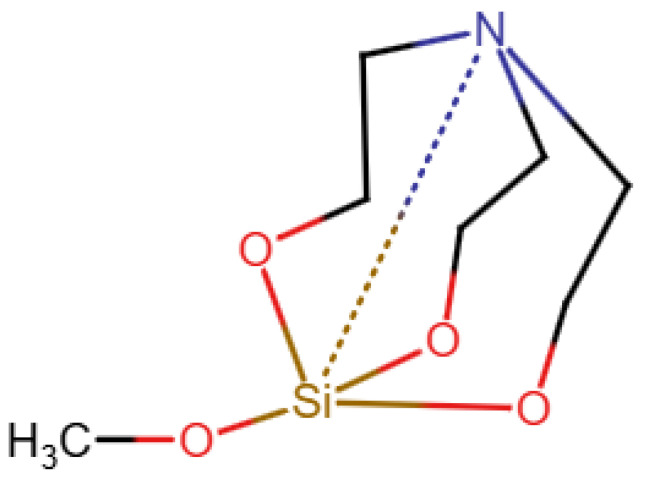
A silicone compound from the demo version of the database in [19] (redrawn).

**Figure 5 molecules-30-03442-f005:**
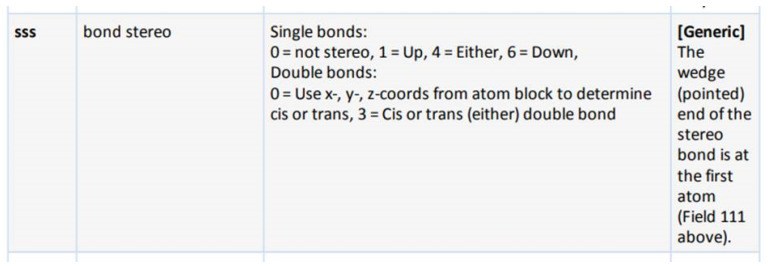
The defintion of wedge bonds in the mol file specification [3] (p. 50).

**Figure 6 molecules-30-03442-f006:**
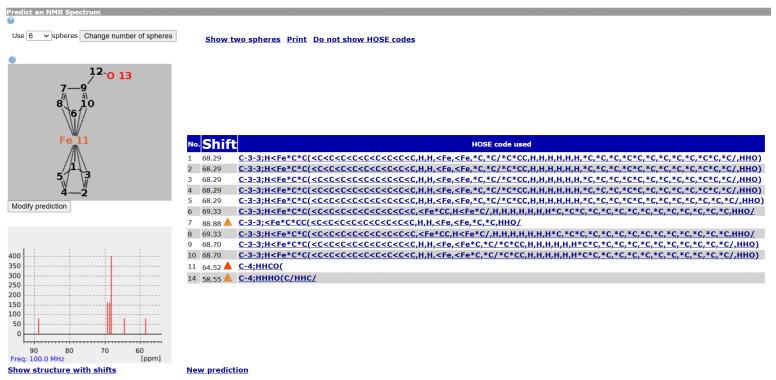
A ^13^*C* NMR prediction for an unknown ferrocene compound.

**Table 2 molecules-30-03442-t002:** Different coordation bonds and how they are depicted using zero-bonds.

Type	Example	Remarks	Example ID in nmrshiftdb2
Metallocenes	See before		80041810
3-centre 2-electron bond	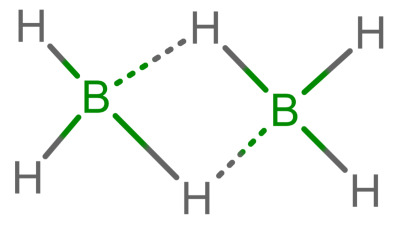	Valences correct Symmetry represented	50120179
Coordinate bridge bond	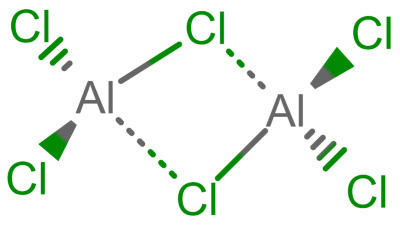	Valences correct Symmetry represented	
Agostic bond	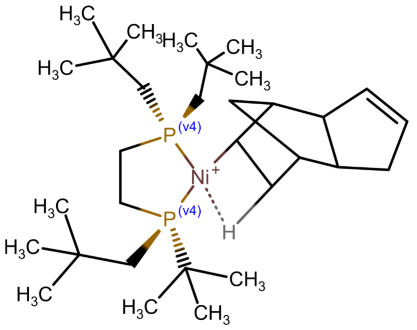	Valences correct All interactions shown	80041816
Coordination complex [7]	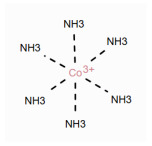	Charges localized All interactions shown	
Hapticity	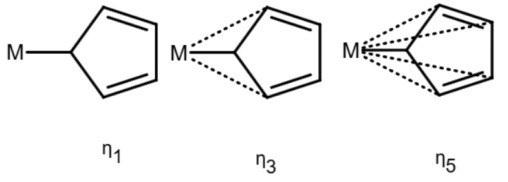	Valences correct All interactions shown	

**Table 3 molecules-30-03442-t003:** Example predictions for organometallic compounds in nmrshiftdb2 using the HOSE code algorithm. * = Warning for low reliability is provided.

Structure	Carbon Number	^13^*C* Shift	^13^*C* Prediction	Error	^1^*H* Shift	^1^*H* Prediction	Error
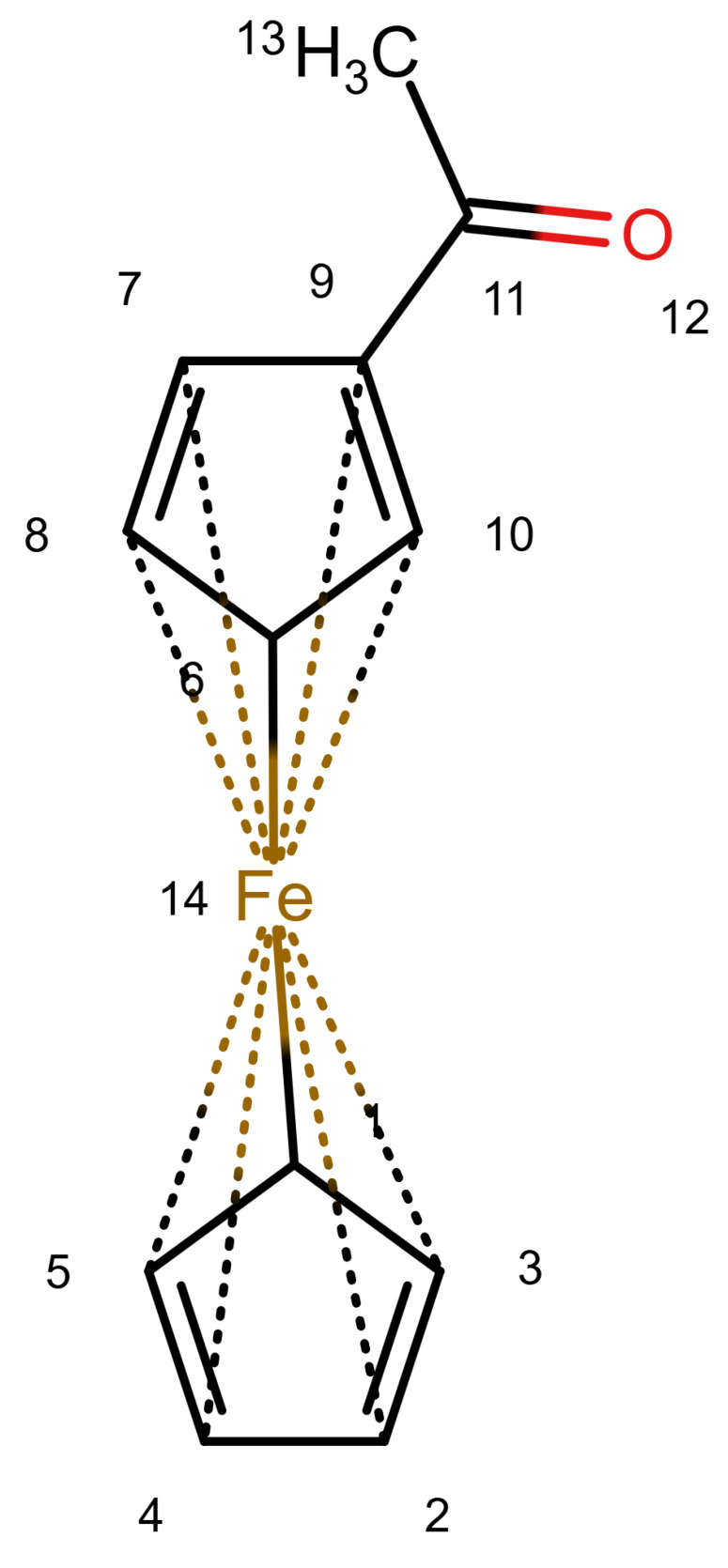	1, 2, 3, 4, 5	69.58	69.36	0.22	4.200	2.75	1.45 *
	7, 10	72.24	69.23	3.01	4.77	2.75	2.02 *
	6, 8	69.79	69.70	0.09	4.5	2.75	1.75 *
	9	79.38	78.61	0.77	-	-	-
	11	201.98	195.31	6.67	-	-	-
	13	27.30	28.61	1.31	2.39	2.33	0.06
	Remark: Predictions are good where similar examples exist (1–5 carbon) and less good in other regions and for hydrogen, where examples are scarce in the database.						
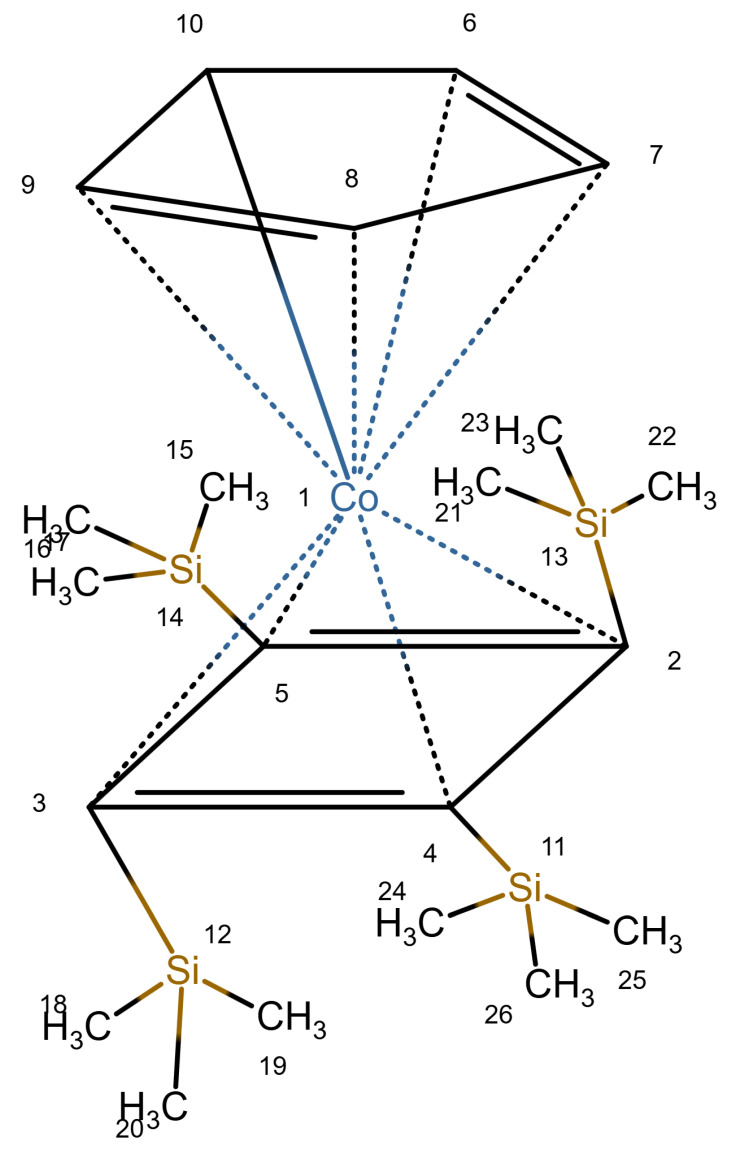	6, 7, 8, 9, 10	79.9	80.01	0.11	5.01	4.95	0.06
	2, 3, 4, 5	82.3	144.36	62.06 *	-	-	-
	6, 8	2.7	0.02	2.68	-	-	-
	Remark: Other cobalt compounds give a good result for 6–10. Since there are no other examples with cobalt and silicon, results for 2–5 are not reliable.						
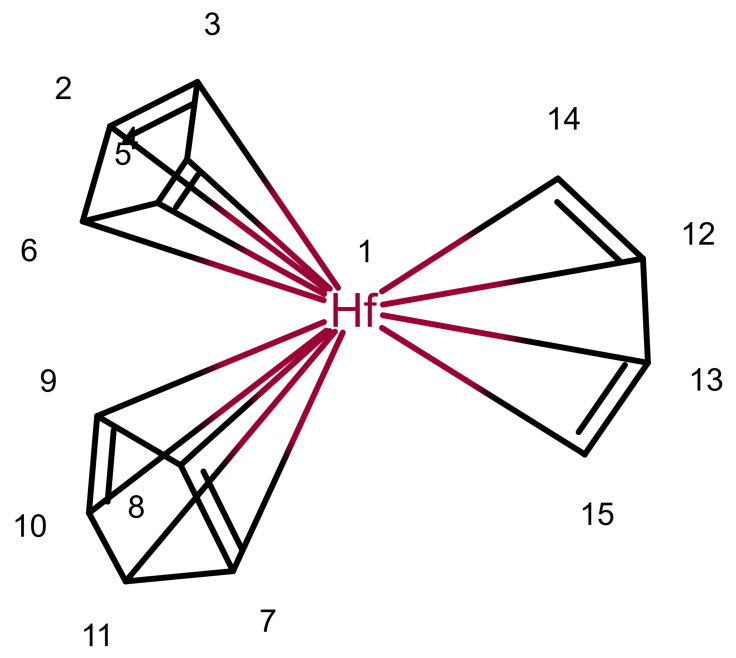	2, 3, 4, 5, 6, 7, 8, 9, 10, 11	102.24	107.36	5.12 *	5.18	6.28	1.10 *
	12, 13	114.58	93.03	21.55 *	4.96	6.95	1.99 *
	14, 15	45.05	77.64	32.59 *	-	-	-
	Remark: No comparable examples exist beyond one sphere, but the first sphere includes the metal, giving roughly comparable results.						

## Data Availability

Data in nmrshiftdb2 can be downloaded from https://sourceforge.net/projects/nmrshiftdb2/files/data/ (accessed on 10 August 2025) and the source code from https://sourceforge.net/p/nmrshiftdb2/code/HEAD/tree/trunk/ (accessed on 10 August 2025). The modified version of the Chemistry Development Kit is available at https://github.com/stefhk3/cdk (accessed on 10 August 2025).

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
