# Peer review of "An NMR Database for Organic and Organometallic Compounds"

_molecules, 2025, doi:10.3390/molecules30163442_

Round 1
Reviewer 1 Report
Comments and Suggestions for Authors
This paper builds on Ref 6 (Clark 2011), which proposed non-organic chemical species could be better represented through the inclusion of zero-order bond and an explicit hydrogen atom count. I am somewhat unclear about the differences in approach here. I note that some key examples in this paper (e.g. ferrocene, at least 1 figure in Table 2) are taken from the original Ref [6] without explicitly noting this.
The title and opening of this paper claim broad applicability to modelling organometallic compounds in computational database systems. However, I find this quite substantial claim to be speculative and insufficiently justified, predominantly due to the lack of in-depth literature review and detailed comparisons to other recent modern cheminformatics literature. Indeed, a communication length would be insufficient for such a paper.
Cheminformatics is a rapidly evolving field, especially given the rapid rise in machine learning in chemistry over the last couple of years; I would expect more recent developments to be considered in this paper and inclusion of a recent review of the topic (perhaps 2022 https://doi.org/10.1002/wcms.1603 that includes as reference Ref 6, but there are very likely others).
Nevertheless, the core result of this paper, implementing the ideas of Ref [6] into nmrshiftdb2 and demonstrating that only minor modifications to the molecular description file are needed, are useful to users of nmrshiftdb2 and perhaps other cheminformatics developers.
For publication, this paper needs a substantial shift in focus to the improvements to the specific software and up-to-date consideration of relevant literature.
Author Response
We thank the reviewer for the helpful comments. We believe we have acted on the comments accordingly. Changes are highlighted in red in the manuscript.
Comment 1: This paper builds on Ref 6 (Clark 2011), which proposed non-organic chemical species could be better represented through the inclusion of zero-order bond and an explicit hydrogen atom count. I am somewhat unclear about the differences in approach here. I note that some key examples in this paper (e.g. ferrocene, at least 1 figure in Table 2) are taken from the original Ref [6] without explicitly noting this.
Response 1: We have added a reference to Clarke to Fig. 1, Table 1 and Table 2 to be absolutely clear. In fact, in Table 1, only two of the examples are shown by Clarke, two of his examples are not in the table. We also make it clear that the whole paper is heavily based on Clarke and his approach. We hope this is clear now.
Comment 2: The title and opening of this paper claim broad applicability to modelling organometallic compounds in computational database systems. However, I find this quite substantial claim to be speculative and insufficiently justified, predominantly due to the lack of in-depth literature review and detailed comparisons to other recent modern cheminformatics literature. Indeed, a communication length would be insufficient for such a paper.
Response 2: We changed title, abstract and conclusion to make clear that we are primarily dealing with NMR data here. We have also extended the background to include more literature. We note that for NMR data the use of zero bonds is novel and there is no specific literature about it. According to https://www.mdpi.com/journal/molecules/instructions#submission molecules only has article and review as contribution types, otherwise we would be happy to make this a communication or similar type.
Comment 3: Cheminformatics is a rapidly evolving field, especially given the rapid rise in machine learning in chemistry over the last couple of years; I would expect more recent developments to be considered in this paper and inclusion of a recent review of the topic (perhaps 2022 https://doi.org/10.1002/wcms.1603 that includes as reference Ref 6, but there are very likely others).
Response 3: We have added more literature (lines 32, 43, 63). As noted, there is not much literature specifically about NMR databases and organometallic compounds. We believe we have cited the relevant items now.
Comment 4: Nevertheless, the core result of this paper, implementing the ideas of Ref [6] into nmrshiftdb2 and demonstrating that only minor modifications to the molecular description file are needed, are useful to users of nmrshiftdb2 and perhaps other cheminformatics developers.
Response 4: This is indeed the core message of the paper. We hope this is clear now.
Comment 5: For publication, this paper needs a substantial shift in focus to the improvements to the specific software and up-to-date consideration of relevant literature.
Response 5: We believe we have done this.
Reviewer 2 Report
Comments and Suggestions for Authors The author attributions are incorrect. They lack information.There are no results or discussion, this complicates the review of the manuscript, the purpose is not clear. The methodology is not clear, it needs to be rethought and be clearer. The conclusions are not adequate, since there is no discussion or results.
All figures shown are of low quality, they should be drawn by the authors to improve quality.
Introduction and Background confuse the reader. You must choose one of the two.
The article's contribution is not clear, the database is not clear. NMR shift values for organometallic compounds are not clear, detail the functions, the program, how the data is calculated.
Author Response
We thank the reviewer for the comments and the opportunity to rephrase our work more clearly. Changes are highlighted in red in the manuscript.
Comment 1: The author attributions are incorrect. They lack information.
Response 1: We have completed the addresses of the authors.
Comment 2: There are no results or discussion, this complicates the review of the manuscript, the purpose is not clear. The methodology is not clear, it needs to be rethought and be clearer. The conclusions are not adequate, since there is no discussion or results.
Response 2: We have added an explicit Results and Discussion section (Section 3, line 266). This makes clear how the conclusions are justified. The methods section should present the method clearly. Overall, this should give a logical flow.
Comment 3: All figures shown are of low quality, they should be drawn by the authors to improve quality.
Response 3: We have redrawn figures wherever possible. They now scale well. For screenshots (Fig. 2, 3, 5, 6) we do not see any option to change it.
Comment 4: Introduction and Background confuse the reader. You must choose one of the two.
Response 4: We have made a single Introduction section, also containing the background.
Comment 5: The article's contribution is not clear, the database is not clear. NMR shift values for organometallic compounds are not clear, detail the functions, the program, how the data is calculated.
Response 5: With the new Results and Discussion section, we hope that the contribution is clear. All values in the database are measured, there is no calculation done for this. We think it is clear from the text that we have an NMR database, which is searchable, also for predictions. We have added in Section 2.4 line 222 an explanation and a link for HOSE codes.
Round 2
Reviewer 2 Report
Comments and Suggestions for Authors
It is clear that the author wishes to highlight his NMR prediction program, however, it is necessary that in the manuscript, he includes an example comparing the actual displacements vs. the predicted ones. This can be done using a table and a figure, to highlight the work he has done and for the reader to understand the author's important application. The author must demonstrate the use of the HOSE code and the NMR predictor with some examples. Explicit examples are suggested in the supplementary material. 1H and 13C NMR must be shown.
Author Response
Comment 1: It is clear that the author wishes to highlight his NMR prediction program, however, it is necessary that in the manuscript, he includes an example comparing the actual displacements vs. the predicted ones. This can be done using a table and a figure, to highlight the work he has done and for the reader to understand the author's important application. The author must demonstrate the use of the HOSE code and the NMR predictor with some examples. Explicit examples are suggested in the supplementary material. 1H and 13C NMR must be shown.
Response 1: We thank the reviewer for this helpful idea. We have added examples on p. 10 of the manuscript. We believe these show a good evaluation of the strength and weaknesses of the current prediction system.